# Atypical Teratoid Rhabdoid Tumor: How Tumor Diagnostic Methods in the Laboratory Have Evolved over the Past 40 Years

**DOI:** 10.3390/cancers17233768

**Published:** 2025-11-26

**Authors:** Heather L. Smith, Pascale Aouad, Nitin R. Wadhwani

**Affiliations:** 1Department of Pathology, Loyola University Medical Center, Maywood, IL 60153, USA; 2Department of Medical Imaging, Ann & Robert H. Lurie Children’s Hospital of Chicago, Feinberg School of Medicine, Northwestern University, Chicago, IL 60611, USA; 3Department of Pathology and Laboratory Medicine, Ann & Robert H. Lurie Children’s Hospital of Chicago, Feinberg School of Medicine, Northwestern University, Chicago, IL 60611, USA

**Keywords:** atypical teratoid/rhabdoid tumor (AT/RT), SMARCB1, SMARCA4, malignant rhabdoid tumor

## Abstract

This opinion article examines the historical evolution of laboratory-based tumor diagnostic methods over the past four decades, with a detailed light microscopic characterization of Atypical Teratoid/Rhabdoid Tumor across both typical and challenging presentations. It also includes time points of importance where the spectrum of immunohistochemical and molecular alterations have changed for the pathologic diagnosis of Atypical Teratoid/Rhabdoid Tumor. It also includes morphologic details of liquid based preparations which can be useful during frozen section analyses and diagnostic lumbar punctures. It highlights the role of molecular testing in clarifying complex cases, discusses potential differential diagnoses based on location and light microscopy morphology, and outlines emerging directions for translational research focused on this aggressive pediatric central nervous system tumor. Diagnostic imaging studies are also included as Atypical Teratoid/Rhabdoid Tumor can sometimes be distinguished from other tumors that are in the differential diagnosis based on the age, location, and imaging characteristics of the tumor.

## 1. Introduction

Malignant Rhabdoid Tumor of the kidney was first described in a small cohort of Wilm’s tumors in the First National Wilm’s Tumor study in 1978. From those 427 cases, 8 had a peculiar rhabdomyosarcomatoid pattern, given the presence of polygonal cells with abundant acidophilic cytoplasm and vesicular nuclei containing a central, prominent nucleolus [1]. Many academic medical centers reported the co-occurrence of these tumors in the kidney and posterior fossa. It was almost a decade later that these tumors were recognized as a separate entity, Malignant Rhabdoid Tumor, after cytogenetic studies identified monosomy 22 as a non-random chromosomal abnormality in these neoplasms [2,3]. Many such cases were documented at national meetings as abstracts and case reports of cerebral Rhabdoid Tumors as early as 1985. When this tumor occurs in the central nervous system, the term Atypical Teratoid/Rhabdoid Tumor (AT/RT) applies, as this was historically regarded as a unique central nervous system tumor [4], often misdiagnosed as Medulloblastoma when it contained a predominant embryonal (“PNET”) appearance. Heterologous histologic elements (squamoid, glandular) and mesenchymal differentiation, while uncommon, were also documented in this large cohort of 32 infants and children.

As pathologists, molecular geneticists, and physician scientists worked to precisely identify this tumor and differentiate it from its clinical, imaging, and histologic mimics, SMARCB1 (INI-1) immunohistochemistry proved to be a valuable diagnostic tool to diagnose AT/RT [5]. A pivotal advancement followed with the discovery of another SWI/SNF chromatin-remodeling complex member associated with Rhabdoid Tumors in infancy—the ATPase subunit SMARCA4 (BRG-1)—leading to the recognition of a second histologic subtype: SMARCA4-mutant AT/RT. Today, the combined use of immunostains INI-1 and BRG-1 remains a powerful surrogate tool for the histologic diagnosis of AT/RT and provides guidance for confirmatory lab testing.

With the advancement of molecular techniques, the histologic diagnosis of AT/RT has been increasingly supported by methods such as next-generation sequencing, deletion/duplication analyses, and tumor microarray testing to detect and confirm SMARCB1 or SMARCA4 alterations. More recently, DNA methylation profiling [6,7] and gene pathway enrichment studies [8,9,10] have enabled further classification of AT/RT into three distinct molecular subgroups: TYR, SHH, and MYC. The ATRT-SHH subgroup features genes that drive overexpression of the Sonic Hedgehog signaling pathway, including GLI2, BOC, PTCHD2, and MYCN. The ATRT-TYR subgroup is characterized by genes upregulating the melanosomal pathway, such as TYR, TYRP, MITF, and OTX2. Lastly, the ATRT-MYC subgroup is defined by overexpression of MYC and genes from the HOX cluster. Identification of these molecular subgroups may lead to novel tumor-directed therapy.

## 2. Incidence

The overwhelming majority of patients diagnosed with AT/RT are under three years of age. In the CBTRUS statistical report, there were 417 cases reported to the registry between 2017 and 2021. A total of 7% of these tumors were diagnosed in the adolescent and young adult (AYA) population and 4% in adults 40 and older [11].

## 3. Localization

In the aforementioned cohort of 32 infants and children, the majority of cases were located in the posterior fossa; however, AT/RT can occur throughout the CNS. Synchronous Malignant Rhabdoid Tumors can occur in the brain and kidney [12,13].

## 4. Imaging-Based Diagnostic Methods

AT/RT can arise anywhere within the CNS, and approximately half of all cases occur within the posterior fossa, with a predilection for an off-midline distribution. The heterogeneous imaging characteristics of AT/RTs reflect their underlying histopathologic complexity [14].

On computed tomography (CT), AT/RTs typically appear hyperdense owing to their high cellularity, and calcifications are frequently identified. Magnetic resonance imaging (MRI) findings are variable on both T1- and T2-weighted sequences; however, restricted diffusion is usually present. Cystic or necrotic components, as well as intratumoral hemorrhage, are common features [15]. Arslanoglu et al. described the presence of eccentrically positioned cysts with peripheral wall enhancement as a potentially distinguishing characteristic of infratentorial AT/RTs (Figure 1). The pattern and degree of contrast enhancement are variable, further reflecting the histopathologic heterogeneity of these neoplasms (Figure 2 and Figure 3). A characteristic pattern of band-like enhancement surrounding a central cystic or necrotic area was observed in 38% of MRIs from a series of 32 patients [16].

Nowak et al. reported distinct MRI features among the three molecular subgroups ATRT-TYR, ATRT-MYC, and ATRT-SHH, including differences in tumor location, cyst morphology, and enhancement patterns. They found that ATRT-TYR tumors occur predominantly in the infratentorial compartment, ATRT-MYC tumors demonstrate a supratentorial predilection, and ATRT-SHH tumors are distributed both supra- and infratentorially. Peripheral cysts are more frequently encountered in the ATRT-TYR subgroup. Additionally, ATRT-TYR and ATRT-MYC tumors tend to exhibit more robust contrast enhancement compared with ATRT-SHH. While these imaging features may assist in subgroup stratification, reliable differentiation among molecular subtypes remains limited by imaging at present [17]. A more recent study found that a considerable subset of MYC ATRTs develop in extra-axial regions, including along the cranial nerves [18].

Imaging features of ATRT may overlap with those of other primitive neuroectodermal tumors, particularly Medulloblastoma in the posterior fossa (15), as the posterior fossa is the most common location for pediatric brain tumors. Utilizing advanced artificial intelligence (AI) algorithms to interpret medical imaging data may lead to tools for tumor segmentation; however, currently the imaging overlap of AT/RT, Medulloblastoma, and other CNS embryonal tumors that can occur in the posterior fossa does not allow for precise stratification within the posterior fossa embryonal tumor category [19].

## 5. Spread

Dissemination of AT/RT follows CSF fluid pathways, and M1 (tumor cells in the CSF) is present at diagnosis in approximately 38% of patients [20] and rarely in patients who require venticuloperitoneal (VP) shunts [21]. Visible nodular seeding by MRI and M2 involvement is potentially more common in spinal AT/RT [22].

## 6. Histopathology

Most cases of AT/RT will be ***histologically defined*** and will be composed of sheets of a high-grade epithelioid neoplasm containing cells with rhabdoid differentiation. These cells are variably sized and contain abundant eosinophilic cytoplasm with the nucleus eccentrically placed in the cell. The nucleus typically has a vesicular appearance and nucleolar hypertrophy (Figure 4A). Some cases may be composed exclusively of a primitive embryonal tumor consisting entirely of hyperchromatic nuclei, and the diagnosis of AT/RT is only apparent after evaluating for INI-1 or BRG-1 immunohistochemistry (Figure 4B). Epithelioid cells may form a trabecular pattern resembling chordoma (Figure 4C) or contain an amyloid-like matrix in the background (Figure 4D). The most challenging cases personally encountered are those with divergent or heterologous differentiation. These tumors may have a papillary architecture and mesenchymal differentiation (Figure 4E,F). This atypical type of morphology can be further diagnostically challenging in older patients. One case in particular contained Schiller–Duval-like bodies, raising the consideration of a yolk sac tumor in the differential diagnosis. This case even stained for SALL4, which serves as a reminder that AT/RT can have a polyphenotypic immunohistochemical pattern of expression not necessarily limited to keratin, muscle, and neuronal immunohistochemical stains.

As all nucleated cells should express INI-1 and BRG-1, surrogate immunohistochemistry is routinely used for light microscopic diagnosis. Loss of normal nuclear expression for either of these immunohistochemical stains should prompt the diagnosis of AT/RT pending additional molecular studies if necessary. Rare cases may show a Golgi-like pattern with peri-nuclear accumulation of INI-1 (or BRG-1) adjacent to the nucleus (Figure 5A,B) or diffuse cytoplasmic localization of the antibody. This cytoplasmic localization has been described previously by Pathak et al., who showed that cytoplasmic localization may be a result of either a single nucleotide variant, small insertions, or deletions that result in a frameshift mutation that alters the protein structure and resulting in either defective protein folding, mislocalization, or abnormal tracking of one of the core subunit proteins of the ATP-dependent SWI/SNF chromatin-remodeling complex [23]. This phenomenon is of potential therapeutic interest, as future trials using nuclear transport inhibitors may be of value for these patients.

On H&E intraoperative stained smears, the tumor will appear as sheets of epithelioid cells with abundant eosinophilic cytoplasm. Occasional slightly larger cells with a plasmacytoid appearance and nucleolar hypertrophy (Rhabdoid cells) should be visible as well (Figure 6A,B). On liquid-based Wright-stained preparations, clusters of large cells with high N:C ratios are usually present (Figure 6C). Occasionally, single large cells with nucleolar hypertrophy are the only abnormality present until CSF dissemination declares itself with time and serial CSF sampling (Figure 6D).

## 7. AT/RT Molecularly Defined

Rare cases of AT/RT will be ***molecularly defined*** as they will have retained expression of INI-1 and BRG-1 [24]. Evaluating a tumor in this setting is broadly accomplished with a combination of next-generation sequencing, deletion and duplication studies, a tumor microarray, and DNA methylation analysis.

A panel utilized for next-generation sequencing should have the ability to detect mutations in both SMARCB1 and SMARCA4. If a mutation is not identified, deletion of the SMARCB1 gene (22q11.2) can be detected by visual inspection of the copy number plots. If the platform is not validated to report on copy number variants, additional testing for deletion or duplication of Chromosome 22q11.23 can be performed using FISH. A tumor microarray may be utilized to either prove a stable genome or unstable genome in the case of a de-differentiated tumor with a rhabdoid appearance.

Prior to the ubiquity of next-generation sequencing, deletion and duplication studies were commonly utilized to confirm the histologic diagnosis of AT/RT. These studies are boarder genomic analyses used to evaluate where DNA segments are missing or repeated. This type of analysis is often used in conjunction with germline testing to evaluate the patient’s overall status and determine if this is a sporadic occurrence of AT/RT or an AT/RT arising in the setting of rhabdoid predisposition syndrome.

DNA methylation has also become a powerful tool to molecularly define AT/RT. This technique essentially analyzes patterns of 5-methylcytosine across the tumor genome to reflect both genetic and epigenetic change in the tumor cells. It can be used to confirm the diagnosis, as the WHO includes methylation studies in the algorithm for essential diagnostic criteria.

Using combined epigenetic studies and gene expression analysis, Ho et al. have shown that the ATRT-TYR subgroup tends to have whole or partial loss of Chromosome 22 with an inactivating point mutation on the other allele. The ATRT-SHH subgroup tends to be enriched for compound heterozygous point mutations. The ATRT-MYC subgroup tends to have broad loss of SMARCB1 [25].

## 8. Differential Considerations for Epithelioid INI-1-Deficient Tumors

The primary diagnostic considerations for isolated, INI-1-deficient tumors in the CNS are epithelioid sarcoma, poorly differentiated chordoma with loss of INI-1, MPNST with loss of INI-1, and Cribriform Neuroepithelial tumor (CRINET). CRINET should be easy to distinguish based on light microscopic evaluation alone, as this tumor is a non-Rhabdoid neuroectodermal tumor with loss of SMARCB1 expression. This tumor was first described in 2009 in a case series of two young children with non-Rhabdoid Tumors around the third and fourth ventricles, showing cribriform and trabecular architecture [26], and since that time has been incorporated as a provisional entity in the [27].

The former differentials raised are more complex, especially if the tumor is disseminated at presentation or exclusively sellar (AT/RT vs. epithelioid sarcoma). These rare tumors have been historically distinguished based on immunophenotype. CD34 immunoreactivity may prompt a diagnosis of proximal-type epithelioid sarcoma, and Brachyury may aid in the morphologic diagnosis for a poorly differentiated chordoma with loss of INI-1, especially with clival involvement. This binary, dogmatic approach may simplify the diagnostic odyssey for these rare tumors; however, DNA methylation can offer another layer of molecular clarity based on epigenetic profiling [28] for sellar-based tumors with rhabdoid differentiation, where it has been shown that sellar-based AR/RT may have high expression of CD34.

Epithelioid Malignant Peripheral Nerve Sheath Tumor can also occur in the CNS, and when it contains loss of INI-1, the primary differential diagnosis is AT/RT. Both tumors can express neural markers (S-100, SOX10) and keratin. De novo Epithelioid MPNST is not associated with NF1, so additional molecular testing may be warranted to differentiate between these two entities. Here, copy number alterations are a useful tool as Epithelioid MPNST has an unstable genome compared with AT/RT. Inactivation of CDKN2A and gain of Chromosome 2q can also be seen in Epithelioid MPNST.

Lastly, choroid plexus carcinoma (CPC) with loss of INI-1 may mimic an AT/RT, as both may present in very young patients. Endoscopic biopsies may exclusively show a poorly differentiated rhabdoid population with loss of INI-1 and overexpression of P53. More extensive resections should demonstrate a papillary epithelial neoplasm with histologic features reminiscent of choroid plexus that has de-differentiated. Occasional pleomorphic, bizarre-appearing cells may also be seen. There have not been enough studies to evaluate the objective use of Kir7.1 to reliably differentiate AT/RT from INI-1-deficient CPC. The overexpression of p53 in this setting should prompt a morphologic diagnosis of malignant neoplasm with loss of INI-1, with a comment stating that additional molecular testing can provide more diagnostic clarity. Here, next-generation sequencing will be the most helpful. A relatively stable genome with a deletion, duplication, or mutation in SMARCB1 is be diagnostic of AT/RT. INI-1-deficient CPC should have a TP53 mutation (or loss of the TP53 region) with an unstable genome containing large-scale gains and losses. Germline testing of the peripheral blood may also be informative and show a TP53 mutation. Previous comparative studies [29] support this diagnostic approach.

It is important to note that SMARCA4 mutations can be observed in Medulloblastoma, which has been increasingly identified with the more ubiquitous use of next-generation sequencing in pediatric brain tumors. Currently, these should be localized to methylation class Medulloblastoma, non-WNT/non-SHH, Group 3 subtype, subclass II [30]. If DNA methylation is not available, it can be beneficial to demonstrate retained BRG-1 expression in the tumor after review of the somatic alterations.

## 9. Extremely Rare Collision/Composite Tumors

There are case studies in the literature that report Malignant Rhabdoid Tumors in families with Neurofibromatosis Type 2 (NF2). The SMARCB1 gene is located on Chromosome 22q11.23, and NF2 is adjacent to Chromosome 22q12.2. Prevailing theories include multiple hits to these loci that promote tumorigenesis [31,32]. In families with NF2 germline alterations, an additional somatic alteration in SMARCB1 may drive the formation of AT/RT without the presence of a germline SMARCB1 abnormality.

## 10. Future Directions

Since the formal identification of AT/RT, diagnostic methods have significantly advanced and have most recently been refined using DNA methylation. The combination of liquid biopsy [33] and evolving epigenetic biomarkers may inform initial imaging and diagnostic studies and may transform surveillance studies in the future. Liquid biopsy may be pre-operatively informative; however, the nature of the tumor still requires urgent neurosurgical intervention. There is also future potential for retrospective neuroimaging AI analyses after molecular characterization of these tumors to potentially segregate embryonal tumors that can occur in the posterior fossa (i.e., Medulloblastoma vs. AT/RT vs. CNS embryonal tumors with PLAGL amplification).

## 11. Concluding Remarks

Atypical Teratoid/Rhabdoid Tumor (AT/RT) is among the most morphologically and biologically distinctive neoplasms of the central nervous system in pediatric populations. However, its diagnosis in adolescent and young adults or adult patients presents greater challenges, often due to age-related diagnostic bias. A standardized, systematic diagnostic approach—incorporating clinical context, radiographic features, histopathologic evaluation, and molecular profiling when necessary—remains essential for identifying even the most diagnostically ambiguous cases.

## Figures and Tables

**Figure 1 cancers-17-03768-f001:**
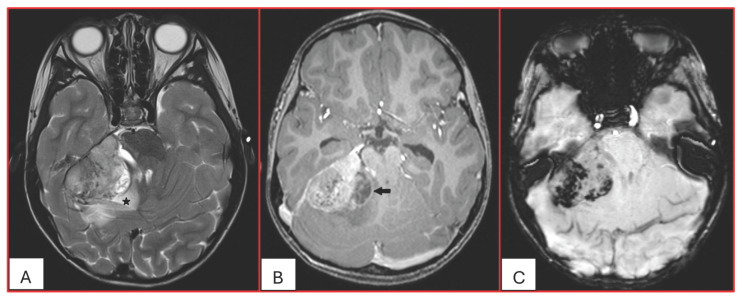
Five-year-old male with ATRT-MYC subgroup. (**A**). Axial T2-weighted image demonstrates a large extra-axial mass in the posterior fossa centered at the right cerebellopontine angle of intermediate heterogeneous signal. There is surrounding vasogenic edema (star) and mass effect. (**B**). Axial post contrast T1-weighted image shows heterogeneous enhancement with eccentric peripherally enhancing cystic component (arrow). (**C**). Axial SWI demonstrates multiple internal foci of susceptibility reflecting hemorrhagic content (no calcifications on the concurrent CT). This tumor was associated with leptomeningeal spread at the time of diagnosis into the internal auditory canals and along the cerebellar folia.

**Figure 2 cancers-17-03768-f002:**
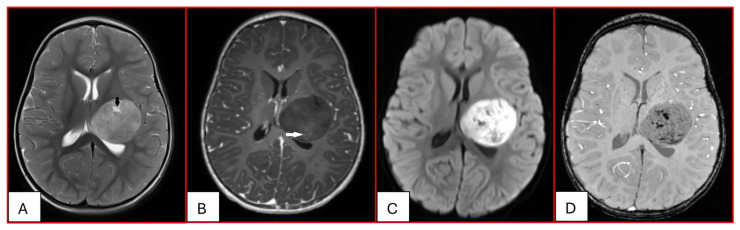
Twenty-three-month-old female with ATRT. (**A**) Axial T2-weighted image reveals a circumscribed supratentorial mass centered in the left deep gray nuclei of heterogeneous intermediate signal with small central cystic/necrotic changes (black arrow). There is no surrounding edema. (**B**) This mass is predominantly non-enhancing with minimal internal foci of enhancement (white arrow). (**C**) There is corresponding prominent restricted diffusion. (**D**) Multiple internal foci of susceptibility are present, which may reflect internal hemorrhage or calcifications.

**Figure 3 cancers-17-03768-f003:**
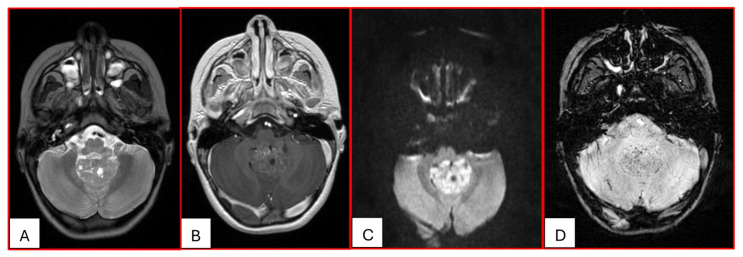
Three-year-old female with ATRT. (**A**) Axial T2-weighted image shows a mass centered in the midline posterior fossa of heterogeneous intermediate signal with small cystic/necrotic changes. There is no surrounding edema. (**B**) This mass demonstrates mild heterogeneous enhancement. (**C**) There is prominent corresponding restricted diffusion. (**D**) A few foci of susceptibility are noted, which may reflect internal hemorrhage or calcifications. Note that the MRI features of this mass mimic the appearance of Medulloblastoma.

**Figure 4 cancers-17-03768-f004:**
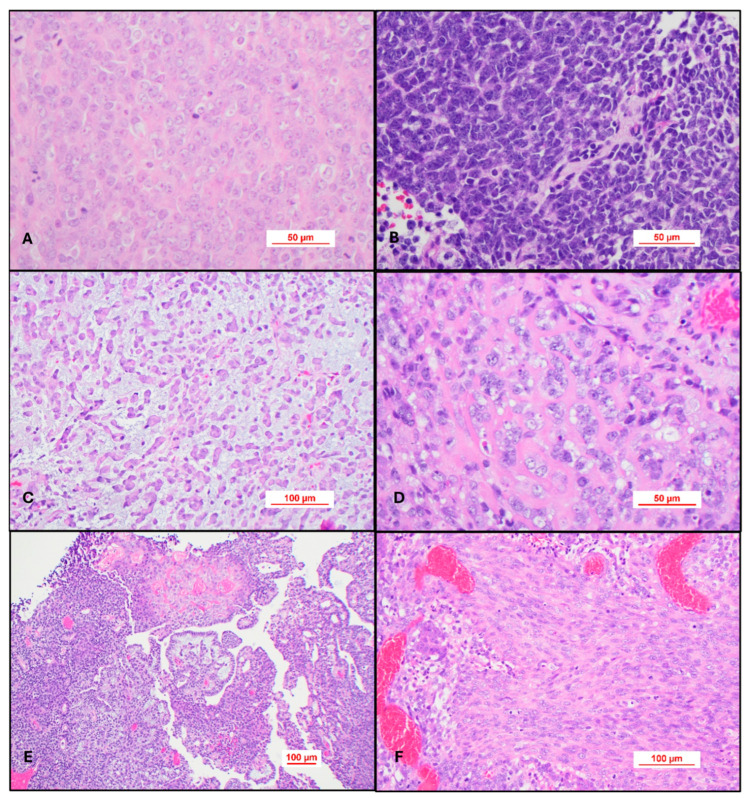
Histopathology of Atypical Teratoid Rhabdoid Tumor. (**A**) H&E sections typically show sheets of variably sized plasmacytoid cells with abundant eosinophilic cytoplasm and a single prominent nucleolus, H&E 40×. (**B**). Some cases may be composed exclusively of a primitive embryonal tumor consisting entirely of hyperchromatic nuclei—Medulloblastoma-like. H&E, 40×. (**C**) Epithelioid cells may form a trabecular pattern resembling chordoma. H&E, 10×. (**D**) Matrix production in the background may appear amyloid-like. H&E, 40×. (**E**) Papillary epithelioid structure resembling a yolk sac tumor. H&E, 10×. **(F**) Both epithelioid and mesenchymal differentiation. H&E, 20×.

**Figure 5 cancers-17-03768-f005:**
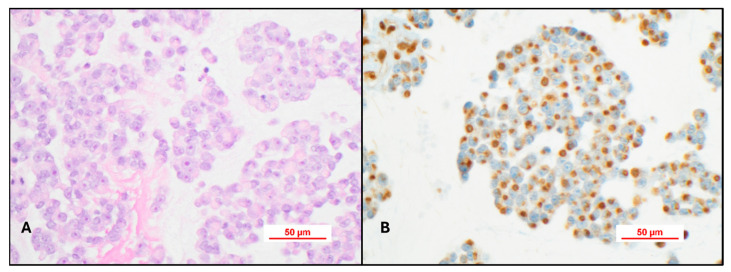
(**A**) H&E section shows classic AT/RT light microscopic features. H&E, 40× (**B**) INI-1 immunohistochemical stain showing a Golgi-like pattern with peri-nuclear accumulation of INI-1. H&E, 40×.

**Figure 6 cancers-17-03768-f006:**
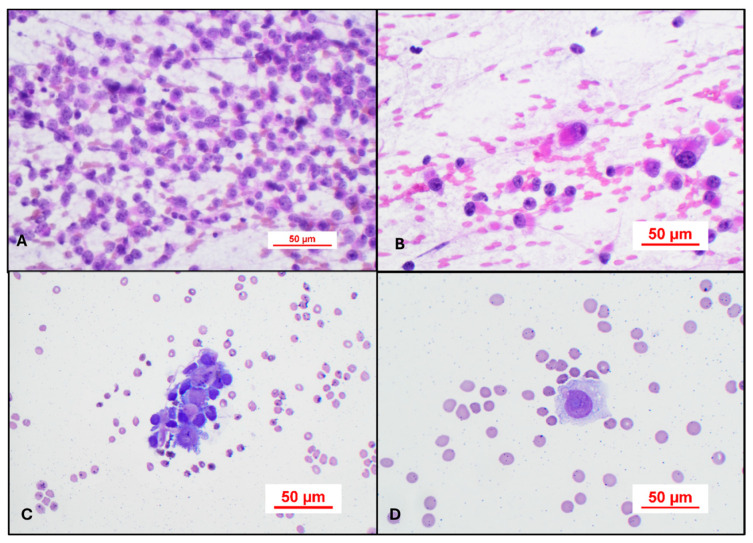
(**A**) H&E intraoperative stained smears demonstrating sheets of epithelioid cells with abundant eosinophilic cytoplasm. H&E, 40×. (**B**) Large Rhabdoid cell. H&E, 40×. (**C**) In liquid-based Wright-stained preparations (40×), clusters of large cells in a case of AT/RT that has CSF dissemination. (**D**) Single large cell (60×) with nucleolar hypertrophy in a case of CSF dissemination prior to detection by MRI.

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
