# Peer review of "Atypical Teratoid Rhabdoid Tumor: How Tumor Diagnostic Methods in the Laboratory Have Evolved over the Past 40 Years"

_cancers, 2025, doi:10.3390/cancers17233768_

Round 1
Reviewer 1 Report
Comments and Suggestions for Authors
The manuscript entitled “Atypical Teratoid Rhabdoid Tumor: How Tumor Diagnostic Methods Have Evolved Over the Past 40 Years” presents an overview of the evolution of diagnostic approaches for Atypical Teratoid/Rhabdoid Tumors (AT/RT) over the past four decades. The topic is important and of considerable interest to both neuropathologists and neuro-oncologists. However, several key aspects in the field are missing, and addressing them would significantly enhance the comprehensiveness and scientific rigor of the manuscript.
Major Comments:
- The authors should include a section detailing imaging-based diagnostic methods (example; MRI, CT, and X-ray etc) in the context of AT/RT. Specific neuroimaging patterns characteristic of AT/RT—such as heterogeneous enhancement, intratumoral hemorrhage, necrosis, and restricted diffusion on MRI—should be discussed. Moreover, studies have indicated potential correlations between MRI/CT features and molecular subgroups of AT/RT. Discussing these associations in detail would provide a more integrated perspective on diagnostic advancements.
- Recent findings, including those reported by the group of Martin Hasselblatt (PMID: 34003336; DOI: 10.1007/s00401-021-02328-w), have described cytoplasmic localization of SMARCB1 in a subset of AT/RT cases, particularly those harboring specific SMARCB1 mutations. These observations have important diagnostic implications and should be discussed in detail. Furthermore, the manuscript would be significantly strengthened by including a schematic representation of the common SMARCB1 mutations identified in AT/RT patient samples, as these alterations have direct relevance to molecular diagnostics.
- It is recommended that the authors include a comprehensive schematic summarizing the full spectrum of diagnostic approaches for AT/RT, including imaging-based techniques and key molecular diagnostic markers associated with this tumor entity.
- The authors are encouraged to elaborate on recent developments in the use of artificial intelligence (AI), machine learning, and radiomics for the automated diagnosis and classification of AT/RT using MRI and CT images. Several recent studies have demonstrated how AI-assisted models can distinguish AT/RT from other pediatric brain tumors by learning radiographic signatures. Incorporating this discussion would highlight the translational potential of computational diagnostics in improving diagnostic precision.
- The authors should expand the future perspectives section to provide their insights on current diagnostic gaps and emerging avenues for improving early and accurate AT/RT detection. For instance, potential directions may include integrating molecular profiling, liquid biopsy approaches (e.g., circulating tumor DNA or exosomal markers), and AI-based predictive models into diagnostic workflows. Articulating a clear vision for the field will make this review more forward-looking and impactful.
- The scale bars are not clearly visible in Figures 1–3. The authors should ensure that scale bars are properly incorporated into each figure and clearly described in the corresponding figure legends, as this is essential for scientific clarity and reproducibility.
Author Response
Reviewer 1
Dear reviewer 1
We are incorporated changes into the manuscript based on your feedback.
A. The authors should include a section detailing imaging-based diagnostic methods (example; MRI, CT, and X-ray etc) in the context of AT/RT. Specific neuroimaging patterns characteristic of AT/RT—such as heterogeneous enhancement, intratumoral hemorrhage, necrosis, and restricted diffusion on MRI—should be discussed. Moreover, studies have indicated potential correlations between MRI/CT features and molecular subgroups of AT/RT. Discussing these associations in detail would provide a more integrated perspective on diagnostic advancements.
Our response: While the manuscript was primarily geared towards discussing the evolution of laboratory based methods, we have added a neuroradiologist as an author and have included an additional component “Imaging based diagnostic methods” with a discussion targeting the commentary suggested above which also includes commentary on the use of AI in neuroimaging (reviewer point 4).
B. Recent findings, including those reported by the group of Martin Hasselblatt (PMID: 34003336; DOI: 10.1007/s00401-021-02328-w), have described cytoplasmic localization of SMARCB1 in a subset of AT/RT cases, particularly those harboring specific SMARCB1 mutations. These observations have important diagnostic implications and should be discussed in detail. Furthermore, the manuscript would be significantly strengthened by including a schematic representation of the common SMARCB1 mutations identified in AT/RT patient samples, as these alterations have direct relevance to molecular diagnostics.
Our response: We have incorporated the findings by Hasselblatt’s group as INI-1 localization is dependent on the genome of the C-terminal region which can potentially inform future targeted therapies preventing nuclear export. We have also added to the portion of the manuscript AT/RT molecularly defined with the studies by Ho et al (PMID 33283872) showing the linkage between the common genomic alterations in different molecular subgroups of AT/RT.
C. It is recommended that the authors include a comprehensive schematic summarizing the full spectrum of diagnostic approaches for AT/RT, including imaging-based techniques and key molecular diagnostic markers associated with this tumor entity.
Our response: We do agree that this addition would make a more robust manuscript for a review article, this is primarily an opinion article on how laboratory methods have changed over time.
D. The authors are encouraged to elaborate on recent developments in the use of artificial intelligence (AI), machine learning, and radiomics for the automated diagnosis and classification of AT/RT using MRI and CT images. Several recent studies have demonstrated how AI-assisted models can distinguish AT/RT from other pediatric brain tumors by learning radiographic signatures. Incorporating this discussion would highlight the translational potential of computational diagnostics in improving diagnostic precision.
Our response: We have added a pediatric neuroradiologist as an author and have included additional commentary related to the use of AI in neuroimaging.
E. The authors should expand the future perspectives section to provide their insights on current diagnostic gaps and emerging avenues for improving early and accurate AT/RT detection. For instance, potential directions may include integrating molecular profiling, liquid biopsy approaches (e.g., circulating tumor DNA or exosomal markers), and AI-based predictive models into diagnostic workflows. Articulating a clear vision for the field will make this review more forward-looking and impactful.
Our response: We added additional potential future directions for growth in the neuroimaging AI space.
F. The scale bars are not clearly visible in Figures 1–3. The authors should ensure that scale bars are properly incorporated into each figure and clearly described in the corresponding figure legends, as this is essential for scientific clarity and reproducibility.
Our response: All figures have been modified.
Reviewer 2 Report
Comments and Suggestions for Authors
In the work, „Atypical Teratoid Rhabdoid Tumor: How tumor diagnostic methods have evolved over the past 40 years. “, the authors Smith et Wadhwani present a short, nice and conclusive review containing some relevant information.
The review is relatively short, containing, for example, in the “epidemiology” section just two sentences regarding age distribution, but no other information.
The title seems to indicate a historical overview since 1985, however, the term “Atypical Teratoid / Rhabdoid Tumor” has only been used since 1995, and in this current paper no year data are given except for the first description of rhabdoid kidney tumor in 1978.
From a historical point of view, it would be interesting to learn, when the different steps were taken: description of rhabdoid tumor, description of AT/RT, identification of the role of INI-1 in immunohistochemistry, and of the genetic alterations in this context.
The fact that there is a possible genetic predisposition for rhabdoid tumors, is not explained in detail. The autosomal dominant heterozygous disease-causing germline variant in SMARCB1 (RTPS1) or SMARCA4 (RTPS2) is an important condition that should be considered and discussed with the families in each case.
In the “spread” section, only the number of 38% cases with csf spread is given; there may be solid metastases as well as just M1 involvement, and primary as well as secondary metastasis, this should be mentioned.
Liquid biopsy in the context of AT/RT is mentioned in the subsection “future directions”. Current state-of-the-art knowledge should be given here.
The sentence “These studies are boarder genomic analyses to evaluate the genome where DNA segments are missing or repeated.” in line 150 seems somewhat corrupted.
Author Response
Dear Reviewer 2:
Thank you for the feedback on the manuscript.
A. The review is relatively short, containing, for example, in the “epidemiology” section just two sentences regarding age distribution, but no other information.
Our response: Epidemiology was a bit broad for this heading. We have modified the heading from Epidemiology to Incidence.
B. The title seems to indicate a historical overview since 1985, however, the term “Atypical Teratoid / Rhabdoid Tumor” has only been used since 1995, and in this current paper no year data are given except for the first description of rhabdoid kidney tumor in 1978.
Our response: We have added additional information to the manuscript as Many cases were documented at national meetings as abstracts and case reports of cerebral rhabdoid tumors as early as 1985.
C. From a historical point of view, it would be interesting to learn, when the different steps were taken: description of rhabdoid tumor, description of AT/RT, identification of the role of INI-1 in immunohistochemistry, and of the genetic alterations in this context.
Our response: We have included in the introduction some time points of critical interest. (1) INI-1 immunohistochemistry, (2) monosomy 22, (3) BRG-1 immunohistochemistry, (4) DNA Methylation.
D. The fact that there is a possible genetic predisposition for rhabdoid tumors, is not explained in detail. The autosomal dominant heterozygous disease-causing germline variant in SMARCB1 (RTPS1) or SMARCA4 (RTPS2) is an important condition that should be considered and discussed with the families in each case.
Our response: While we agree with your opinion that RTPS should be discussed in a review article, this is primarily written from the laboratory diagnostic point of view as it relates to the tumor alone.
E. In the “spread” section, only the number of 38% cases with csf spread is given; there may be solid metastases as well as just M1 involvement, and primary as well as secondary metastasis, this should be mentioned.
Our response: We have expanded on this section to include M1 and M2 metastasis with appropriate citations.
F. Liquid biopsy in the context of AT/RT is mentioned in the subsection “future directions”. Current state-of-the-art knowledge should be given here.
Our response: We added additional commentary on liquid biopsy in this section.
G. The sentence “These studies are boarder genomic analyses to evaluate the genome where DNA segments are missing or repeated.” in line 150 seems somewhat corrupted.
Our response: We have re-phrased this sentence
Round 2
Reviewer 1 Report
Comments and Suggestions for Authors
The authors were able to clearly address most of the comments raised in this study, and the suggested modifications have been taken into account in improving the quality of the article. Authors were also able to incorporate suggested additional sections along with high-resolution images in the revised version of the manuscript. The revised version of the manuscript is clear, concise, and well-written. I recommend this article for publication.
Reviewer 2 Report
Comments and Suggestions for Authors
My concerns have been met.